# Bilateral Idiopathic Neuroretinitis

**DOI:** 10.3390/diagnostics14212386

**Published:** 2024-10-26

**Authors:** Cosmin Adrian Teodoru, Horațiu Dura, Mihai Dan Roman, Adrian Hașegan, Ciprian Tănăsescu, Andrei Moisin, Doina Ileana Giurgiu, Mihaela Laura Vică, Horia Stanca, Maria-Emilia Cerghedean-Florea, Corina Suteu

**Affiliations:** 1Faculty of Medicine, “Lucian Blaga” University of Sibiu, 550024 Sibiu, Romania; 2Department of Cellular and Molecular Biology, “Iuliu Haţieganu” University of Medicine and Pharmacy, 400012 Cluj-Napoca, Romania; 3Institute of Legal Medicine, 400006 Cluj-Napoca, Romania; 4Department of Ophthalmology, “Carol Davila” University of Medicine and Pharmacy, 050474 Bucharest, Romania; 5Faculty of Medicine and Pharmacy, University of Oradea, 410081 Oradea, Romania

**Keywords:** neuroretinitis, macular fan, optic disc edema

## Abstract

Background/Objectives: Neuroretinitis (NR) is a rare inflammatory condition characterized by sudden vision loss, optic disc edema and macular star appearance predominantly affecting individuals in their third and fourth decades of life. Methods: This paper describes the case of 33-year-old Caucasian man with no significant medical history complaining of decreased vision for about a week. Results: The ophthalmological exam revealed best-corrected visual acuity (BCVA) for the right eye (RE) of 0.8 (decimal notation) and of 0.9 for the left eye (LE). Intraocular pressure (IOP) was 20 mmHg in RE and 18 mmHg in LE. Slit-lamp examination of both eyes (OU) showed no evidence of intraocular inflammation in the anterior chamber or vitreous cavity. Examination of the posterior pole of the right eye showed bilateral papilledema with an incomplete macular fan pattern. Conclusions: Despite extensive laboratory tests, including serologic and imaging investigations, a definitive etiology remained unclear. It is very important to differentiate NR from other optic nerve disorders, requiring careful clinical evaluation and observation of the evolution of symptoms.

Neuroretinitis (NR) is a rare inflammatory disease that typically presents with sudden visual loss, optic disc edema and lipid exudates in a macular star appearance [1]. It mainly occurs in the third and fourth decades of life, mostly unilaterally [2,3]. The clinical presentation of unilateral neuroretinitis includes impaired visual acuity, dyschromatopsia, relative afferent pupillary defects and visual field abnormalities [4]. In our case, the eye examination showed bilateral papilledema, more pronounced in the right eye, with the presence of an incomplete macular fan (Figure 1).

The pathophysiologic mechanism of NR involves increased permeability of the disk vasculature, causing exudation of fluid into the peripapillary retina [2]. Kitamei et al. showed that that a single arteriole on the disc surface was the source of a significant dye leak, rather than to the optic disc capillaries’ extensive leakage. Therefore, the lipid-rich fluid circulates directly into the outer nuclear-plexiform space to accumulate under the neurosensory retina, forming a star-shaped pattern due to the radiating configuration [5].In our case, optical coherence tomography (OCT) confirmed ophthalmoscopic findings by showing optic disc edema (OU) and neurosensory macular detachment in the RE (Figure 2).

The sudden decrease in visual acuity in young people is unusual, unlike in older people, where a number of pathologies such as cataracts [6], glaucoma [7,8], macular pathology [9] or other factors can be incriminated [10] for the slow or sudden decrease in visual acuity. In certain situations, genetic testing has an extremely important role in establishing a diagnosis [11,12]. Neuroretinitis is thought to closely resemble unilateral optic disk edema disorders [13]. Diagnosis is difficult, especially at an early stage, when classic features may not always be present [14]. Typically, one of the three features of neuroretinitis are visual field defects. The most common field defect is caecocentral scotoma, but central scotomas, arcuate defects and even altitudinal defects may also be present [13]. In our case, visual field was performed at presentation (Figure 3) and one month later (Figure 4).

Numerous infectious diseases, including syphilis [15], toxoplasmosis [16], tuberculosis [17], varicella [18], herpes simplex virus [19], herpes zoster virus [20], mumps [21] and cat scratch disease have been linked to neuroretinitis [22]. In rare occasions, certain inflammatory diseases such as sarcoidosis, systemic lupus erythematosus, Behcet’s disease, polyarteritis nodosa, Takayasu arteritis, Vogt–Koyanagi–Harada disease or inflammatory bowel disease have been associated with neuroretinitis [4]. In some cases, neuroretinitis may have an atypical presentation [23,24,25,26]. Sometimes, non-arteritic anterior ischemic anterior ischemic optic neuropathy (NAAION) may also present with optic disc edema and hard exudates that extend into the peripapillary retina, mimicking neuroretinitis, which makes the diagnosis even more difficult [27]. However, it seems that, unfortunately, fifty percent of the cases have no identifiable cause and are labelled as idiopathic neuroretinitis [28]. Vision loss in neuroretinitis is usually unilateral, but in rare cases it can be bilateral [29]. It can range from 20/20 to light perception and tends to be more severe in cat scratch neuroretinitis (CSD-NR) and idiopathic recurrent neuroretinitis [2].

In our case, the neurological examination was normal. Arterial blood pressure was within normal limits (120/80 mmHg at first presentation) and was stable during the entire follow-up period. In some cases, systemic hypertension can cause vision loss through retinopathy, choroidopathy and optic neuropathy. However, before diagnosing hypertensive papillary edema, it is essential to rule out various causes of optic disc edema [30,31]. In this case, the results of the laboratory tests, including complete blood count, erythrocyte sedimentation rate, C-reactive protein, fibrinogen, antinuclear antibody, perinuclear antineutrophil cytoplasmic antibodies and angiotensin-converting enzyme, were within normal limits at the time. Kidney and liver function were within normal limits. Serologic tests showed cytomegalovirus (CMV) IgM negative and CMV IgG positive with a value of 7.92 UI/mL (normal values 0–1 UI/mL). Also, serologic tests for Bartonella, toxoplasma, hepatitis B, hepatitis C, human immunodeficiency virus, herpes simplex, varicella zoster, Epstein–Barr, borrelia, rubella and syphilis were negative. Chest X-ray was normal, and tuberculin skin test was also negative. Magnetic resonance imaging (MRI) performed 7 days after the presentation was normal. In this case, the etiology was not confirmed despite extensive evaluation. The patient received intravenous methylprednisolone (1 g/day) for 3 days, followed by prednisolone 1 mg/kg/day, with a progressive decrease in doses over the next weeks.

One month later, BCVA was 0.9 in the right eye (RE) and 1 in the left eye (LE). A decrease in macular edema (RE) and a slight improvement in visual field (OU) (Figure 4) were noticed. Six weeks after the first presentation, the optic nerve swelling had decreased but exudates persisted in the peripapillary region (Figure 5). Treatment options of neuroretinitis include combination therapy, steroids and antibiotics, depending on the underlying cause. It is usually a self-limiting disease. With or without treatment, most patients have an excellent recovery of vision [32]. Normally, the optic nerve will heal in 6–8 weeks, with manifestation of a normal or slightly pale optic disk. Meanwhile the macula exudates will resolve more slowly, starting several to 6–12 weeks [33,34,35]. However, some cases have been reported in the literature without visual improvement due to optic disk atrophy [36]. Low visual acuity has a crucial impact on quality of life, especially in young patients [37]. Steroid pulse therapy was used in this case as the first line of treatment when infectious reasons were ruled out. An important aspect to highlight is the asymmetry of the disease in the left eye. A plausible explanation could be the start of corticosteroid treatment, although there is some evidence that corticosteroids are more effective when used in combination with specific antibiotics [32].

It is important to differentiate neuroretinitis from a number of conditions with a similar fundus appearance, such as common forms of optic neuropathies including papillary edema, ischemic optic neuropathy, optic neuritis, compressive lesions, some toxic or nutritional deficiencies as well as hereditary forms [38,39]. Extensive laboratory tests, including serologic and imaging investigations, are essential to identify potential infectious or inflammatory causes [40].

To conclude, neuroretinitis is a complex and often challenging condition that causes severe visual impairment due to inflammation of the optic nerve and retina. The rarity of the disease, together with the overlap of its symptoms with other optic nerve disorders, highlights the importance of extensive clinical investigation in these cases. An accurate diagnosis is essential not only for an effective treatment, but also to rule out serious underlying conditions that may require different therapeutic approaches. The management of this pathology, particularly idiopathic cases, highlights the importance of a multidisciplinary approach. Treatment schedules, such as corticosteroids, can lead to improvements in visual outcomes, but these need to be adjusted based on individual patient assessments and continuous monitoring.

## Figures and Tables

**Figure 1 diagnostics-14-02386-f001:**
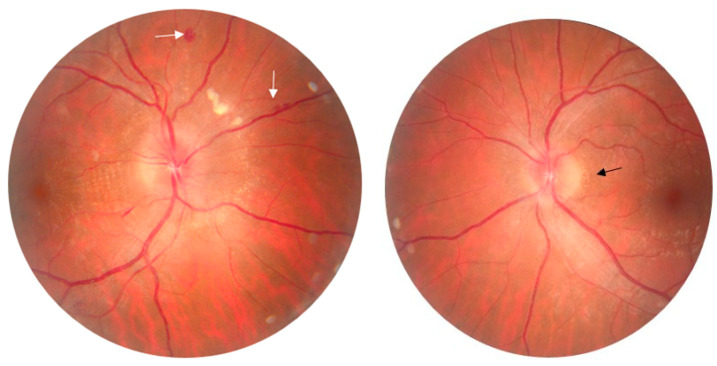
Retinophotography of both eyes (OU) in a 33-year-old Caucasian man with no significant medical history presenting to our service with decreased visual acuity for about a week. There were no other constitutional symptoms, history of tuberculosis or contact with dogs, cats or other pets. The ophthalmological examination revealed best-corrected visual acuity (BCVA) in the right eye (RE) of 0.8 and 0.9 for the left eye (LE). Intraocular pressure (IOP) was 20 mmHg in RE and 18 mmHg in LE. Slit-lamp examination (OU) showed no evidence of intraocular inflammation in the anterior chamber or vitreous cavity. Color vision was impaired in the right eye more than that in the left eye. There was no pain on ocular movement. Examination of the posterior pole (RE) shows an elevated optic nerve papilla with blurred contour, a few peripapillary microhemorrhages (white arrow) and a cotton wool spot, as well as hard exudates, characteristically arranged in the pattern of an incomplete macular fan. In the LE we noticed a slightly elevated optic nerve papilla with blurred contour, a few exudates in the peripapillary region (black arrow) and some perivascular microhemorrhages in the infero-temporal sector.

**Figure 2 diagnostics-14-02386-f002:**
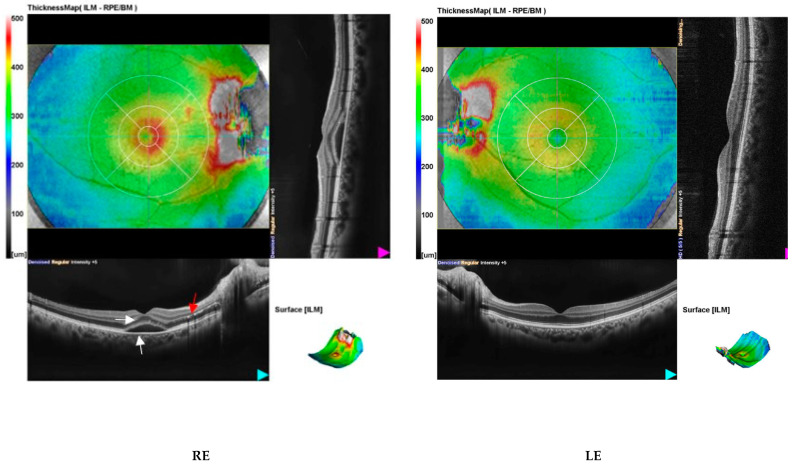
Optical coherence tomography (OCT) of the OU at first presentation showed bilateral papilledema with peripapillary subretinal fluid presence. In the RE, we noticed the presence of intraretinal and subretinal fluid (white arrow) extending from the optic disc along with neurosensory macular detachment and hard exudates in the inner layers of the retina (red arrow). Central macular thickness was 429 μm in the right eye and 317 μm in the left eye.

**Figure 3 diagnostics-14-02386-f003:**
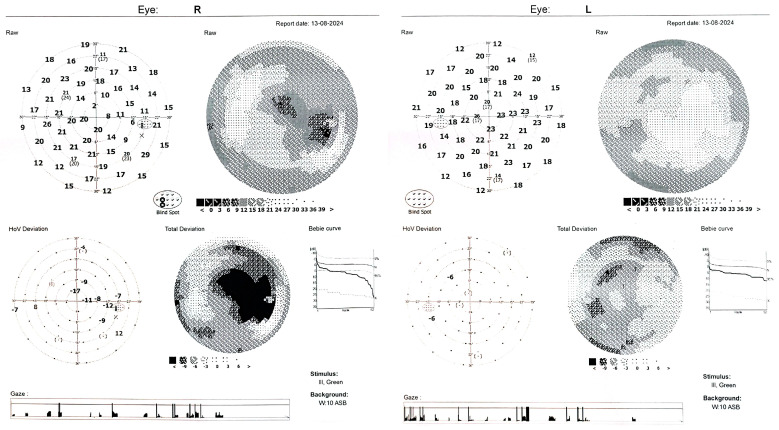
Visual field of both eyes performed at first presentation. A decrease in retinal sensitivity (OU) was noticed. In the RE, the major defect is a superior macular arcuate scotoma and a large blind spot (RE). The other eye (LE) shows similar arcuate defects but is much less dense.

**Figure 4 diagnostics-14-02386-f004:**
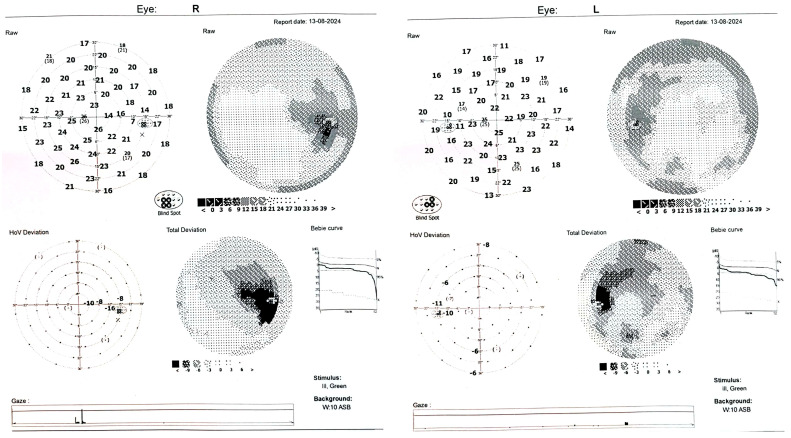
Visual field of both eyes performed after one month follow-up. The patient’s clinical features improved after the treatment. BCVA was 0.9 in RE and 1 in LE. Intraocular pressure (IOP) was 15 mmHg in RE and 16 mmHg in LE.

**Figure 5 diagnostics-14-02386-f005:**
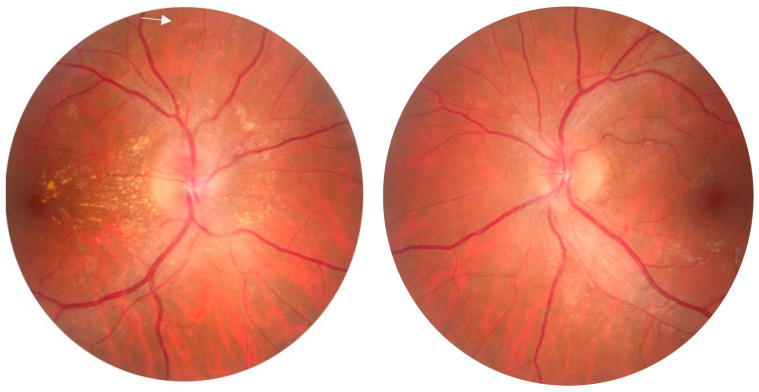
Retinophotography of both eyes (OU) at six weeks after the first presentation. We noticed that the optic nerve swelling decreased in both eyes and visual acuity was back to normal but exudates persisted in the peripapillary region along with a few peripapillary microhemorrhages (RE).

## Data Availability

The data presented in this study are available on request from the corresponding author.

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
