# Peer review of "Bilateral Idiopathic Neuroretinitis"

_diagnostics, 2024, doi:10.3390/diagnostics14212386_

Round 1
Reviewer 1 Report
Comments and Suggestions for Authors
1. Abstract: in line 24, “Examination of the posterior pole of the right eye showed bilateral papillary edema with an incomplete macular star 24 pattern in the right eye.” “of the right eye” should be deleted
2. Figure 2 description: the macular OCT of RE shows a neurosensory macular detachment, not an RPE detachment (line 52).
3. Third paragraph of Discussion: in lines 76-78, the end of the sentence should be rewritten as follows: “….inflammatory bowel disease that have been associated with neuroretinitis in rare cases.” Or: “In rare occasions certain inflammatory diseases such as sarcoidosis, systemic lupus erythematosus, Behcet's disease, polyarteritis nodosa, Takayasu arteritis, Vogt-Koyanagi-Harada disease, inflammatory bowel disease have been associated with neuroretinitis.”
4. Line 83: “…range from 20/20 to light perception.”
5. Line 85: “…and the rest of the cranial nerves..” is redundant, as well as “no meningeal signs”. Neurological examination includes cranial nerve evaluation and possible meningeal signs.
6. Figure 4: One cotton-wool nodules are visualized in each eye, and are not mentioned, despite a vascular condition could be at the origin of this presumed neuroretinitis case. Fluorescein angiography could have been of great help in the evaluation of the retinal and choroidal circulatory status.
7. No serology for Bartonella is informed.
8. No reference to blood pressure evaluation: this is extremely important, as severe systemic arterial hypertension, especially those cases associated with kidney disease, that may be ignored by the patient.
9. Lines 117-122: High blood pressure is not mentioned, and in all cases of bilateral optic disc edema with uni or bilateral lipid deposits it should be rouled out, as this association needs urgent evaluation. For this purpose, references about this association should be included.
10. Poor quality of fundus images
Comments on the Quality of English LanguageImprove phrasing.
Author Response
Thank you for your appreciations and recommendations.
Below are our responses to your comments.
- Abstract: in line 24, “Examination of the posterior pole of the right eye showed bilateral papillary edema with an incomplete macular star 24 pattern in the right eye.” “of the right eye” should be deleted
Response: The text was corrected.
- Figure 2description: the macular OCT of RE shows a neurosensory macular detachment, not an RPE detachment (line 52).
Response: The text was corrected.
- Third paragraph of Discussion: in lines 76-78, the end of the sentence should be rewritten as follows: “….inflammatory bowel disease that have been associated with neuroretinitis in rare cases.” Or: “In rare occasions certain inflammatory diseases such as sarcoidosis, systemic lupus erythematosus, Behcet's disease, polyarteritis nodosa, Takayasu arteritis, Vogt-Koyanagi-Harada disease, inflammatory bowel disease have been associated with neuroretinitis.”
Response: The text was corrected.
- Line 83: “…range from 20/20 tolight perception.”
Response: The text was corrected as recommended.
- Line 85: “…and the rest of the cranial nerves..” is redundant, as well as “no meningeal signs”. Neurological examination includes cranial nerve evaluation and possible meningeal signs.
Response: The text has been reworded for better understanding
- Figure 4: One cotton-wool nodules are visualized in each eye, and are not mentioned, despite a vascular condition could be at the origin of this presumed neuroretinitis case. Fluorescein angiography could have been of great help in the evaluation of the retinal and choroidal circulatory status.
Response: We improved the quality of the photos in the manuscript and also completed and reformulated the description of them. Unfortunately, we could not perform angiography with fluorescein, although very helpful... it is not available in our clinic.
- No serology for Bartonella is informed.
Response: Serologic tests for bartonella were negative, the information was added to the manuscript
- No reference to blood pressure evaluation: this is extremely important, as severe systemic arterial hypertension, especially those cases associated with kidney disease, that may be ignored by the patient.
Response: Blood pressure has always been within normal parameters. The information has been added to the manuscript. We have also added information about kidney function.
- Lines 117-122: High blood pressure is not mentioned, and in all cases of bilateral optic disc edema with uni or bilateral lipid deposits it should be rouled out, as this association needs urgent evaluation. For this purpose, references about this association should be included.
Response: Blood pressure has always been within normal parameters. The information has been added to the manuscript. We have also added information discussion paragraph related to this aspect
- Poor quality of fundus images
Response: We have improved the quality of the photos in the manuscript and also completed and reformulated the description of them.
Comments on the Quality of English Language
Improve phrasing.
Response: We checked and improved our English
Reviewer 2 Report
Comments and Suggestions for Authors
In abstract section, line 23-24, kindly rephrase.
At presentation , could you please comment on the status of the vitreous inflammation in this patient .
In figure 2, the OCT of the left eye does not show any fluid collection in the given picture .
In the right eye could you please highlight/mark the presence of the intra retinal fluid .
In figure 5, macular star can be seen only in the right eye.
Comments on the Quality of English LanguageThe authors could rephrase a few sentences for better understanding of the paper .
Author Response
Thank you for your appreciations and recommendations.
Below are our responses to your comments.
In abstract section, line 23-24, kindly rephrase.
Response: The text was corrected.
At presentation , could you please comment on the status of the vitreous inflammation in this patient .
Response: Information about the status of the vitreous at presentation was added to the manuscript.
In figure 2, the OCT of the left eye does not show any fluid collection in the given picture .
Response: Fluid collection was noticed only in the right eye
In the right eye could you please highlight/mark the presence of the intra retinal fluid .
Response: Yes. Description of the pictures has been improved
In figure 5, macular star can be seen only in the right eye.
Response: Indeed, the characteristic pattern, although incomplete, could only be visualized in the right eye. However, in the left eye (Figure 2), several exudates were seen in the peripapillary region (Figure 2, black arrow).
Comments on the Quality of English Language
The authors could rephrase a few sentences for better understanding of the paper
Response: We checked and improved our English
Reviewer 3 Report
Comments and Suggestions for Authors
The authors present an interesting case of bilateral simultaneous neuroretinitis. I have attached an annotated copy of the PDF. An unusual feature is that only the right eye shows subretinal fluid. The authors must account for this. One possibility an asymmetry in onset, with the right eye more advanced than the left, and the process was cut short in the left eye by the corticosteroids. If correct this is an interesting observation. The authors should look at the recently published classification of optic neuritis Petzold, A., Fraser, C. L., Abegg, M., Alroughani, R., Alshowaeir, D., Alvarenga, R., ... & Plant, G. T. (2022). Diagnosis and classification of optic neuritis. The Lancet Neurology, 21(12), 1120-1134. In which Petzold et al. review neuro-retinitis suggesting that this should be classified as a pre-laminar optic neuritis because the optic nerve is normal on MRI. If no MRI was carried out here this should be mentioned. Also of interest would be the recent paper on anterior ischaemic neuropathy Chapelle, A. C., Rakic, J. M., & Plant, G. T. (2023). The occurrence of intraretinal and subretinal fluid in anterior ischemic optic neuropathy: pathogenesis, prognosis, and treatment. Ophthalmology, 130(11), 1191-1200. in which Chapelle et al. discuss the differences between anterior ischaemic optic neuropathy, where subretinal fluid is common but exudates rare and Neuro-retinitis where exudates are common, indeed required for the diagnosis. The hypothesis is that the former is a transudate and the latter an exudate. Please find attached the annotated PDF of the article

Please see annotated PDF
Author Response
Thank you for your appreciation and recommendations. The manuscript has been revised, and image quality has been improved. As for specific comments related to the text, these have been corrected and are reflected in the revised version of the manuscript. We have also inserted several paragraphs in the discussion part related to the comments that were addressed.
Round 2
Reviewer 1 Report
Comments and Suggestions for Authors
All the suggestions have been addressed adequately
Reviewer 3 Report
Comments and Suggestions for Authors
Please see PDF with minor comments
